# A Combinatorial Perspective on the Optimization of Shallow ReLU Networks

**Michael Matena**
Department of Computer Science
University of North Carolina at Chapel Hill
Chapel Hill, NC 27599
mmatena@cs.unc.edu

**Colin Raffel**
Department of Computer Science
University of North Carolina at Chapel Hill
Chapel Hill, NC 27599
craffel@cs.unc.edu

## Abstract

The NP-hard problem of optimizing a shallow ReLU network can be characterized as a combinatorial search over each training example's activation pattern followed by a constrained convex problem given a fixed set of activation patterns. We explore the implications of this combinatorial aspect of ReLU optimization in this work. We show that it can be naturally modeled via a geometric and combinatoric object known as a zonotope with its vertex set isomorphic to the set of feasible activation patterns. This assists in analysis and provides a foundation for further research. We demonstrate its usefulness when we explore the sensitivity of the optimal loss to perturbations of the training data. Later we discuss methods of zonotope vertex selection and its relevance to optimization. Overparameterization assists in training by making a randomly chosen vertex more likely to contain a good solution. We then introduce a novel polynomial-time vertex selection procedure that provably picks a vertex containing the global optimum using only double the minimum number of parameters required to fit the data. We further introduce a local greedy search heuristic over zonotope vertices and demonstrate that it outperforms gradient descent on underparameterized problems.

## 1 Introduction

Neural networks have become commonplace in a variety of applications. They are typically trained to minimize a loss on a given dataset of labeled examples using a variant of stochastic gradient descent. However, our theoretical knowledge of neural networks and their training lags behind their practical developments.

Single-layer ReLU networks are an appealing subject for theoretical study. The universal approximation theorem guarantees their expressive power while their relative simplicity makes analysis tractable (Hornik, 1991). We restrict ourselves in this paper to studying empirical risk minimization (ERM) as was done in previous works (Du et al., 2018; Oymak & Soltanolkotabi, 2020), which is justified since the train set performance tends to upper bound the test set performance. Furthermore, modern neural networks achieve zero training loss but nevertheless generalize well (Kaplan et al., 2020; Nakkiran et al., 2021). Minimizing the training loss of a shallow ReLU network is a nonconvex optimization problem. Finding its global minima is difficult and can in fact be shown to be NP-hard in general (Goel et al., 2020). Arora et al. (2016) provide an explicit algorithm for finding the global minima by solving a set of convex optimization problems; however, the size of this set is exponential in both the input dimension $d$ and the number of hidden units $m$.

In this paper, we explore the combinatorial structure implicit in the global optimization algorithm of Arora et al. (2016). We start by using tools from polyhedral geometry to characterize the set of convex optimization problems and describe the relationships between the subproblems. Notably, we are

36th Conference on Neural Information Processing Systems (NeurIPS 2022).

able to create a special type of polytope called a zonotope (McMullen, 1971) whose vertices have a one-to-one correspondence with the convex subproblems and whose faces represent information about their relationships. We then explore the combinatorial optimization problem implicit in shallow ReLU network empirical risk minimization using the zonotope formalism to help interpret our findings and assist in some proofs.

Since the computational complexity of optimization problems shapes our approach to solving them, we examine the reductions of NP-hard problems introduced in Goel et al. (2020). The datasets produced have examples that are not in general position (i.e. they have nontrivial affine dependencies), which differs from most real-world datasets. We prove that the global optimum of the loss of a shallow ReLU network over such a dataset can have a discontinuous jump for arbitrarily small perturbations of the data, which has a very natural interpretation in our zonotope formalism. This means that a proof of the NP-hardness of ReLU optimization given training examples in general position does not follow from the results Goel et al. (2020) via a simple continuity argument. We therefore present a modification of their proof that uses a dataset in general position.

In contrast to the NP-hardness of general ReLU optimization, sufficient overparameterization allows gradient descent to provably converge to a global optimum in polynomial time, as demonstrated by (Du et al., 2018; Oymak & Soltanolkotabi, 2020). We interpret the proof methods generally used in these works as asserting that sufficient overparameterization allows gradient descent to bypass much of the combinatorial search over zonotope vertices by having a randomly chosen vertex be close to one with a good solution with high probability. We then introduce a novel algorithm that finds a good zonotope vertex in polynomial time requiring only about twice the minimum number of hidden units required to fit the dataset.

Finally, we explore how gradient descent interacts with this combinatorial structure. We provide empirical evidence that it can perform some aspects of combinatorial search but present an informal argument that it is suboptimal. We reinforce this claim by showing that a greedy local search heuristic over the vertices of the zonotope outperforms gradient descent on some toy synthetic problems and simplifications of real-world tasks. In contrast to the NP-hard worst case, these results suggest that the combinatorial searches encountered in practice might be relatively tractable.

We summarize our contributions as follows.

- We are the first to provide an in-depth exposition of the combinatorial structure arising from the set of feasible activation patterns that is implicit in shallow ReLU network optimization. In particular, we show that this structure can be characterized exactly as a Cartesian power of the zonotope generated by the set of training examples.
- We use this formalism to prove necessary conditions for the global optimum of a shallow ReLU network to be discontinuous with respect to the training dataset. We show that this implies that previous NP-hardness proofs of ReLU optimization do not automatically apply to datasets satisfying realistic assumptions, which we rectify by presenting a modification that uses a dataset in general position.
- We explore the role that combinatorial considerations play in the relationship between overparameterization and optimization difficulty. In particular, we introduce a novel polynomial-time algorithm fitting a generic dataset using twice the minimum number of parameters needed.
- We introduce a novel heuristic algorithm that performs a greedy search along edges of a zonotope and show that it outperforms gradient descent on some toy datasets.

We hope that the tools we introduce are generally useful in furthering our understanding of ReLU networks. Notably, they have deep connections to several well-established areas of mathematics (McMullen, 1971; Richter-Gebert & Ziegler, 2017; Ziegler, 2012), which might allow researchers to quickly make new insights by drawing upon existing results in those fields.

## 2 Empirical Risk Minimization for ReLU Networks

A single ReLU layer consists of an affine transformation followed by a coordinate-wise application of the ReLU nonlinearity $\phi(x) = \max\{x, 0\}$. We can represent an affine transformation from $\mathbb{R}^d \to \mathbb{R}^m$ by an $m \times (d + 1)$ matrix by representing its inputs in homogeneous coordinates, i.e. by appending a $(d + 1)$-th coordinate to network inputs that is always equal to 1. Hence a single ReLU layer with parameters $W$ can be written as $f_W(\mathbf{x}) = \phi(W\bar{\mathbf{x}})$, where $\bar{\mathbf{x}}$ denotes $\mathbf{x}$ expressed

in homogeneous coordinates. A single hidden layer ReLU network consists of a single ReLU layer followed by an affine transformation. We focus on the case of a network with scalar output, so the second layer can be represented by a vector $\mathbf{v} \in \mathbb{R}^{m+1}$. Although the second layer parameters are trained jointly with the first layer in practice, we assume that they are fixed for our analysis. This parallels simplifying assumptions made in previous work (Du et al., 2018).

Let $\ell : \mathbb{R} \times \mathbb{R} \to \mathbb{R}$ be a convex loss function such as MSE or cross-entropy. Since the second layer parameters are fixed, we can incorporate them into a modified loss function $\tilde{\ell} : \mathbb{R}^m \times \mathbb{R} \to \mathbb{R}$ given by $\tilde{\ell}(\mathbf{z}, y) = \ell(\mathbf{v}^T \bar{\mathbf{z}}, y)$ that operates directly on the first layer's activations $\mathbf{z}$. This modified loss function is convex since it is the composition of a convex function with an affine function.

Suppose we are given $\mathcal{D} = \{(\mathbf{x}_i, y_i)\}_{i=1}^N$ as the training dataset with $\mathbf{x}_i \in \mathbb{R}^d$ and $y_i \in \mathbb{R}$ for all $i = 1, \ldots, N$. Throughout this paper, we assume that $N > d + 1$. Sometimes we will represent a dataset by a matrix $X \in \mathbb{R}^{(d+1) \times N}$ with each column corresponding to an example and its labels as the vector $\mathbf{y} \in \mathbb{R}^N$. We say that $\mathcal{D}$ is in general position if there exist no nontrivial affine dependencies between the columns of $X$. The empirical loss $L(W)$, also known as the empirical risk, is defined as the mean per-example loss

$$L(W) = \frac{1}{N} \sum_{i=1}^N \tilde{\ell}(f_W(\mathbf{x}_i), y_i). \tag{1}$$

The goal of ERM is to find a set of parameters $W$ that minimizes this loss. Arora et al. (2016) were the first to introduce an algorithm for exact ERM. We adapt their algorithm for the case of fixed second layer weights in algorithm 1, which has a running time of $O(N^{md} \operatorname{poly}(N, m, d))$. The algorithm works by iterating over all feasible activation patterns

$$\mathcal{A} = \left\{ \mathbb{I}\{W\bar{X} > 0\} \in \{0, 1\}^{m \times N} \mid W \in \mathbb{R}^{m \times (d+1)} \right\}. \tag{2}$$

The subset of parameters corresponding to an activation pattern, which we call an activation region, can be expressed via a set of linear inequalities. Within a single activation region, the map from parameter values to ReLU layer activations over the training dataset is linear. Hence we can solve a constrained convex optimization problem to get the optimal parameters in each activation region. Namely for a given $A \in \mathcal{A}$, we solve for $W \in \mathbb{R}^{m \times (d+1)}$ in the following

$$\begin{aligned} \text{minimize} \quad & \frac{1}{N} \sum_{i=1}^N \tilde{\ell}(\mathbf{a}_i \odot (W\bar{\mathbf{x}}_i), y_i) \\ \text{subject to} \quad & (2\mathbf{a}_i - 1) \odot (W\bar{\mathbf{x}}_i) \geq 0 \end{aligned} \tag{3}$$

where $\odot$ denotes the Hadamard product and $\mathbf{a}_i$ denotes the $i$-th column of $A$. The global optimum then becomes the best optimum found over the entire set of activation regions. Thus single-layer ReLU network ERM can be expressed as a combinatorial search over activation patterns with a convex optimization step per pattern.

## 3 Zonotope Formalism

While Arora et al. (2016) mention that $\mathcal{A}$ arises from a set of hyperplanes induced by the training examples, they only use this connection to bound its cardinality $|\mathcal{A}| = O(N^{md})$. However, hyperplane arrangements are well-studied geometric and combinatoric objects (Richter-Gebert & Ziegler, 2017; Stanley et al., 2004). As such, we will see that we can use this connection to better characterize the combinatorial aspects of ReLU optimization.

Our mathematical tools for describing the combinatorial structure of shallow ReLU network optimization include oriented hyperplane arrangements, polyhedral sets, polyhedral complexes, and zonotopes. Appendix A provides an approachable overview of these topics for unfamiliar readers.

**Single Hidden Unit** We start by considering a single ReLU unit $f_{\mathbf{w}}(\mathbf{x}) = \phi(\mathbf{w}^T \bar{\mathbf{x}})$ parameterized by the vector $\mathbf{w} \in \mathbb{R}^{d+1}$. Looking at its behavior as $\mathbf{w}$ ranges over $\mathbb{R}^{d+1}$ on a single training example $\mathbf{x}_i$, we see that there are two linear regimes depending on the sign of $\mathbf{w}^T \bar{\mathbf{x}}$. They are separated by the hyperplane in parameter space satisfying $\mathbf{w}^T \bar{\mathbf{x}} = 0$. We can describe such behavior mathematically as an oriented hyperplane with the sign of $\mathbf{w}^T \bar{\mathbf{x}}$ providing its orientation. The

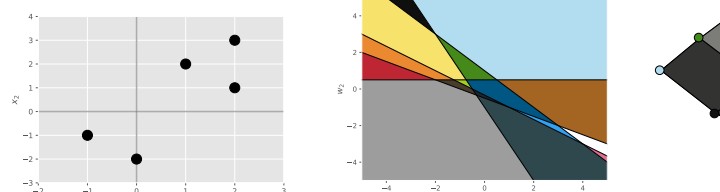

Figure 1: Left: A set of 5 training examples in $\mathbb{R}^2$. Center: A two dimensional slice of parameter space $\mathbb{R}^3$ along the $w_3 = 1$ plane reflecting the polyhedral complex $\mathcal{R}$. The lines correspond to the set of hyperplanes $H_1^0, \ldots, H_5^0$. The different shaded chambers correspond to different activation regions. Each chamber can be thought of as the base of cone whose apex is the origin. Right: The zonotope $\mathcal{Z}$ for this dataset. The corresponding activation region for a vertex is indicated by the colored circles. Note how the edges and faces of $\mathcal{Z}$ capture the incidence structure of the activation regions. Each of the red lines is a translation of a (homogenized) training example. Exactly these 3 training examples are active in the yellow activation region.

collection of oriented hyperplanes associated to each training example $\{\bar{\mathbf{x}}_i\}_{i=1}^N$ is known as an oriented hyperplane arrangement (Richter-Gebert & Ziegler, 2017).

The structure imposed on parameter space $\mathbb{R}^{d+1}$ by this oriented hyperplane arrangement can be described as a polyhedral complex (Ziegler, 2012), which is a collection of polyhedral sets and their faces that fit together in a "nice" way. The polyhedral complex $\mathcal{R}$ induced by the training set will contain codimension 0 sets called chambers. These correspond exactly to activation regions. Activation patterns have a one-to-one correspondence with the tuple of hyperplane orientations associated to each chamber. The center panel of fig. 1 provides an illustration of $\mathcal{R}$ for an example dataset.

---

**Algorithm 1** Exact ERM (Arora et al., 2016)

---

**Input:** data $\mathcal{D} = \{\mathbf{x}_i, y_i\}_{i=1}^N$, 2nd layer $\mathbf{v} \in \mathbb{R}^{m+1}$
$\mathcal{A} \subseteq \{0,1\}^{m \times N}$ {feasible activation patterns (2)}
$W^* \in \mathbb{R}^{m \times (d+1)}$ {random initialization}
**for** $A \in \mathcal{A}$ **do**
    $W \leftarrow$ solution of (3)
    **if** $L(\tilde{W}) < L(W^*)$ **then**
        $W^* \leftarrow W$
    **end if**
**end for**
**return** $W^*$

---

The dual zonotope of a polyhedral complex is a single polytope providing an alternate representation of its combinatorial structure (Ziegler, 2012). Each dimension $k$ member of the polyhedral complex has a corresponding codimension $k$ face in the dual zonotope. Incidence relations between the members of the complex are preserved in the dual zonotope. Generally, a zonotope can be described as the image of an $N$-dimensional hypercube under a linear map whose columns are known as its generators (McMullen, 1971). Each vertex of the zonotope thus is a weighted sum of its generators with coefficients belonging to $\{0, 1\}$.

The dual zonotope $\mathcal{Z}$ of our polyhedral complex $\mathcal{R}$ has the training examples $\{\bar{\mathbf{x}}_i\}_{i=1}^N$ as its generators. The vertices of $\mathcal{Z}$ have a one-to-one correspondence to the activation regions of our network. When a vertex is expressed as a weighted sum over the generators, the coefficient $\{0, 1\}$ of each generator equals its corresponding example's value in the region's activation pattern. The right panel of fig. 1 shows an example zonotope $\mathcal{Z}$ and its duality with the polyhedral complex $\mathcal{R}$.

These correspondences allow us to assign additional structure to the set of activation patterns $\mathcal{A}$ rather than just treating it as an unstructured set. For example, the 1-skeleton of the zonotope $\mathcal{Z}$, which is the graph formed by its vertices and edges, provides a means of traversing the set of activation patterns. Furthermore, we can directly make connections between the training dataset and the activation pattern structure by making use of the fact that $\mathcal{Z}$ is generated by the training examples.

**Multiple Hidden Units** In the multiple hidden unit setting, i.e. $m > 1$, note that parameter space becomes an $m$-fold Cartesian product of single-unit parameter spaces. Furthermore, we are free to set the activation pattern for each unit independently of the others. As the combinatorial structure for

each hidden unit can be described using the zonotope $\mathcal{Z}$, the combinatorial structure for a multiple hidden unit network is described by the $m$-fold Cartesian product $\mathcal{Z}^m = \prod_{i=1}^m \mathcal{Z}$. As noted in appendix A.3.1, $\mathcal{Z}^m$ is also a zonotope. Each vertex of $\mathcal{Z}^m$ corresponds to a product of $m$ vertices of $\mathcal{Z}$. As in the single unit case, there is a one-to-one correspondence between the vertices of $\mathcal{Z}^m$ and the set of activation patterns $\mathcal{A}$.

# 4    ReLU Optimization

## 4.1    NP-Hardness

Given the additional structure we have imposed on the set of activation patterns in algorithm 1, it is natural to ask whether we can use it to develop a global optimization algorithm that is more efficient than a brute-force search over activation patterns. Unfortunately, several works (Goel et al., 2020; Froese et al., 2021) have demonstrated that global optimization of a shallow ReLU network is NP-hard. Nevertheless, this does not preclude the existence of an efficient combinatorial optimizer given certain conditions on the input dataset. Since the zonotope $\mathcal{Z}^m$ encapsulates the combinatorial structure of the optimization problem, we look to see if properties of $\mathcal{Z}^m$ can be related to the difficulty of combinatorial optimization.

Nontrivial affine dependencies between training examples influence the combinatorial structure of the $\mathcal{Z}^m$. Since the reductions of NP-hard problems to ReLU optimization done in Goel et al. (2020); Froese et al. (2021) create datasets with such nontrivial dependencies, it is natural to ask whether it is NP-hard to optimize a shallow ReLU network over a training dataset in general position.

### 4.1.1    Discontinuity of the Global Optimum

If the global minimum of the loss is always continuous with respect to the input dataset, then the NP-hardness of optimization over arbitrary datasets in general position would follow from continuity since every set of points is arbitrarily close to a set in general position. However, we can prove that such continuity holds unconditionally for ReLU optimization only in the case where the training dataset is in general position. We give a sketch of the proof here along with some analysis of the failure cases that can happen when the data are not in general positions. We provide a full proof in appendix C.

**Theorem 4.1.** *Suppose we are given a dataset $\mathcal{D} = \{(\mathbf{x}_i, y_i)\}_{i=1}^N$ in general position and some $m \in \mathbb{N}$. Let $L^*(\mathcal{D})$ denote the global minimum of the loss* (1) *over the dataset $\mathcal{D}$ for a shallow ReLU network with $m$ units. Given any $\epsilon > 0$, some $\delta > 0$ exists such $|L^*(\mathcal{D}) - L^*(\mathcal{D}_\epsilon)| < \delta$ for any dataset $\mathcal{D}_\epsilon = \{(\mathbf{x}_i', y_i)\}_{i=1}^N$ satisfying $\|\mathbf{x}_i - \mathbf{x}_i'\|_2 \le \epsilon$.*

*Proof sketch.* For a small enough perturbation, we can prove that the datasets' zonotopes are combinatorially equivalent. Hence their sets of feasible activation patterns will be exactly the same. Using the fact that any subset of a set in general position is also in general position, we can then show that the constrained convex optimization problem associated with each vertex is continuous with respect to the input dataset. Since the global minimum of the loss is just the minimum of the optimal loss for each vertex, its continuity follows from the fact that the composition of two continuous functions is continuous. $\qquad\square$

When the dataset $\mathcal{D}$ is not in general position, there are two possible ways in which breaking of nontrivial affine dependencies between examples can cause the global minimum of the loss to become discontinuous. The first is that the globally optimal vertex in the perturbed zonotope exists in the original zonotope, but its associated constrained convex optimization problem is discontinuous with respect to the dataset. This can happen when there are nontrivial affine dependencies that get broken amongst the active examples in the vertex. The second way is that the globally optimal vertex of the perturbed zonotope does not exist in the original zonotope. Geometrically, we can think of such a vertex as resulting from the breakdown of a non-parallelepiped higher dimension face (Gover, 2014). See appendix D for examples of these phenomena.

**Analysis of Reductions**    We can use this characterization of the instabilities of the global optimum to perturbations in the training data to analyze the reductions of NP-hard problems used in Goel et al. (2020). We focus on the reduction of the NP-hard set cover problem to the optimization of

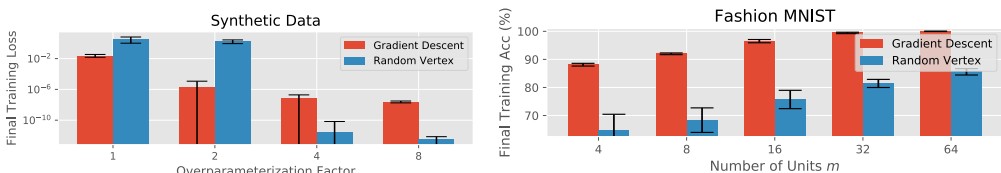

Figure 2: Comparison between gradient descent and optimization with a fixed random activation pattern. Left: Results for MSE on synthetic data for $d = 8$ and $m_{\text{gen}} = 8$. The overparameterization factor times $m_{\text{gen}}$ equals the number of units in the trained network. Right: Results for accuracy on Fashion MNIST coat/pullover binary classification for $d = 16$ and $N = 700$.

a single bias-free ReLU. In appendix E.2, we provide an explicit example of an arbitrarily small perturbation that results in the global minimum of the loss being independent of the solution to the set cover problem.

To the best of our knowledge, existing reductions of NP-hard problems to ReLU optimization all create datasets that are not in general position (Goel et al., 2020; Froese et al., 2021). Therefore, we present a modification of the set cover reduction that produces a dataset in general position. See appendix E.3 for details of this modification along with a proof that it is indeed a reduction of the set cover problem. We thus have the following statement.

**Theorem 4.2.** *Optimizing a ReLU is NP-hard even when restricted to datasets in general position.*

### 4.2 Polynomial Time Optimization via Overparameterization

Even though ReLU network optimization is NP-hard in general, it can be shown that overparameterization allows for gradient descent to converge to the global minimum in polynomial time (Du et al., 2018; Zou & Gu, 2019; Oymak & Soltanolkotabi, 2020; Allen-Zhu et al., 2019). This is not a contradiction since optimization of overparameterized ReLU networks is a strict subset of the set of all ReLU optimization problems.

The general proof method of these works usually involves demonstrating that overparameterization results in activation patterns not changing much throughout training. This allows gradient descent to effectively bypass the combinatorial search of the outer loop in algorithm 1. The remaining optimization problem can then be shown to be similar to the constrained convex optimization problem (3) by assuming that the second layer is frozen. Using the zonotope formalism, we can interpret these results as saying that a sufficiently large number of hidden units $m$ guarantees with high probability that a randomly chosen vertex corresponds to a region of parameter space containing a global minimum of the loss. Parameter initialization selects the random vertex in practice.

This can be justified theoretically through a connection to random feature models. Here we assume that the first layer is frozen, and the second layer forms a linear model over the random first layer features. As the number of units $m$ increases past the number of training examples $N$, the set of first layer activations can become linearly independent. The probability of this approaches 1 as $m \to \infty$. Whether all of the parameters within an activation region produce linearly dependent activations can be shown to depend solely on its activation pattern when the dataset is in general position.

To test this, we ran experiments comparing batch gradient descent to solving (3) for a randomly chosen vertex on some toy datasets. We created synthetic datasets by first choosing the input dimension $d$ and a positive integer $m_{\text{gen}}$. To get the training examples, we sampled $N = (d+1)m_{\text{gen}}$ points in $\mathbb{R}^d$ i.i.d. from the standard Gaussian distribution. We then sampled the weights of a shallow ReLU network with $m_{\text{gen}}$ units i.i.d. from the standard Gaussian distribution. We used this network to create the labels for our synthetic dataset. See appendix H.1.1 for details on the data generation process.

We also created toy binary classification datasets from MNIST (LeCun et al., 2010) and Fashion MNIST (Xiao et al., 2017) by choosing two classes, 5/9 and coat/pullover, respectively, to differentiate. We used the first $d \in \{8, 16\}$ components of the PCA whitened data and selected $N \in \{350, 700\}$ examples for our training sets. See appendix H.2 for details.

We present some of our results in fig. 2. See appendix H for details of the training procedures and for results on more $d, m_{\text{gen}}$ and $d, N$ pairs. On synthetic data, we see that the random vertex method finds a good solution for overparameterization factors of 4 and up. However, gradient descent tends

to arrive at reasonably good solutions for lower levels of overparameterization while the random vertex method fails. This was a general trend that we observed across different $d, m_{\text{gen}}$ pairs on the synthetic datasets and $d, N$ pairs on the binary classification datasets. Note that the Fashion MNIST networks represented in fig. 2 were relatively underparameterized with the maximal size of 64 units being overparameterized by only a factor of about 1.5.

This demonstrates that gradient descent can perform some aspects of the combinatorial search over zonotope vertices. We hypothesize that the gradient tends to be smaller within activation regions with a good optimum and thus gradient descent is more likely to stay within a good activation region. Conversely, the larger gradients within activation regions with poor optima make it more likely that a gradient descent step will move the parameters out of those regions. We can thus think of gradient descent as performing a pseudo-annealing process over the vertices of the zonotope since the likelihood of moving from one vertex to another decreases as the parameters settle into better activation regions.

### 4.2.1 Tighter Bounds

We now introduce a novel vertex selection scheme that runs in polynomial time and requires minimal overparameterization. Suppose $\mathcal{D} = \{(\mathbf{x}_i, y_i)\}_{i=1}^N$ is a dataset in general position. Assume that the examples are ordered by the value of their last coordinate, which we suppose is unique WLOG (i.e. $\mathbf{e}_d^T \mathbf{x}_i < \mathbf{e}_d^T \mathbf{x}_j$ for $i < j$). If not provided in this format, this can be accomplished in $O(N \log N)$ time. We now split the dataset into $\lceil \frac{N}{d+1} \rceil$ chunks containing at most $d + 1$ examples. We write each chunk as $\mathcal{D}_k = \{(\mathbf{x}_i, y_i)\}_{i=(k-1)(d+1)+1}^{\min(N, k(d+1))}$. Since each $\mathcal{D}_k$ contains a contiguous chunk of examples sorted along an axis in coordinate space, we see that we can always find a hyperplane separating $\mathcal{D}_k$ and $\mathcal{D}_{k'}$ for $k \neq k'$. For each $k = 1, \ldots, \lceil \frac{N}{d+1} \rceil$, we add two units to our ReLU network and assign them the activation pattern of 0 for examples belong to a $\mathcal{D}_{k'}$ with $k' < k$ and 1 for the remaining examples. One of the units will be multiplied by +1 in the second layer while the other will be multiplied by -1. Hence the network contains a total of $2\lceil \frac{N}{d+1} \rceil$ hidden units. We prove in appendix F that a set of weights with that activation pattern exists such that the output of the network on training examples exactly matches their labels. The key idea in the proof is that we can sequentially fit the examples in the $k$-th chunk without undoing our progress in fitting the chunks before it.

**Theorem 4.3.** *Given a dataset in $\mathbb{R}^d$ containing $N$ examples in general position, a shallow ReLU network containing $2\lceil \frac{N}{d+1} \rceil$ hidden units can be found in polynomial time exactly fitting the dataset.*

To the best of our knowledge, this is the tightest known bound on the amount of overparameterization needed to find the global optimum of a ReLU network in polynomial time. A simple argument comparing the number of unknowns and the number of equations demonstrates that we need at least $\frac{N}{d+1}$ hidden units to exactly fit an arbitrary dataset with a shallow ReLU network. Hence our method uses only about twice as many hidden units as is necessary to fit the data. However, we emphasize that this ReLU optimization scheme is primarily of theoretical interest since we find in practice that the resulting ReLU network tends to be a very ill-conditioned function.

### 4.3 Relevance to Optimization in Practice

Practically all optimization of ReLU networks in practice uses some variant of gradient descent with an overparameterized network. As the degree of overparameterization goes down, gradient descent begins to arrive at increasingly suboptimal solutions (Nakkiran et al., 2021).

In section 4.2, we hypothesized how gradient descent can find activation regions containing good optima. However, the gradient of the loss is inherently a local property in parameter space while the space's decomposition into activation regions is inherently global. Boundaries between regions correspond to discontinuities in the gradient of the loss. We hypothesize that these properties lead to little direct information about the optimization problem being used to inform gradient descent's traversal over zonotope vertices. Hence we suspect that algorithms that explicitly traverse zonotope vertices using some loss-based criteria can outperform gradient descent in the underparameterized- to mildly-overparameterized regimes.

**Algorithm 2** Greedy Local Search (GLS) Heuristic
***

**Input:** data $\mathcal{D} = \{\mathbf{x}_i, y_i\}_{i=1}^N$, 2nd layer
$\mathbf{v} \in \mathbb{R}^{m+1}$, max steps $T \in \mathbb{N}$
$A_0 \in \text{vert}(\mathcal{Z}^m)$
**for** $t \in \{0, \ldots, T\}$ **do**
    $A_{t+1} \leftarrow A_t$
    **for** $A' \in \text{neighbors}(A_t)$ **do**
        **if** $\mathcal{L}^*(A'; \mathcal{D}) < \mathcal{L}^*(A_{t+1}; \mathcal{D})$ **then**
            $A_{t+1} \leftarrow A'$
        **end if**
    **end for**
    **if** $A_{t+1} = A_t$ **then**
        **return** $A_t$
    **end if**
**end for**
**return** $A_T$
***

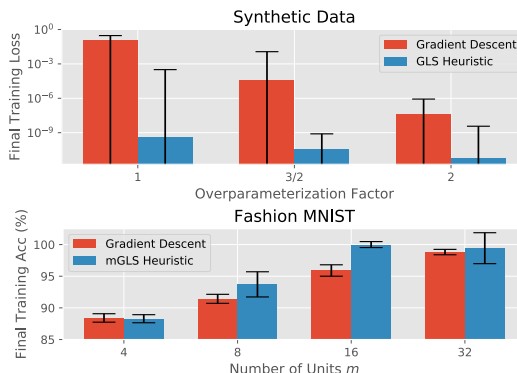

Figure 3: Comparison between gradient descent and our GLS heuristics. Top: Results for MSE on synthetic data for $d = 4$ and $m_{\text{gen}} = 2$. The overparameterization factor times $m_{\text{gen}}$ equals the number of units in the trained network. Bottom: Results for accuracy on Fashion MNIST coat/pullover binary classification for $d = 8$ and $N = 350$.

### 4.3.1 Difficulty of Combinatorial Search

Unless $P = NP$, we are unlikely to find an efficient algorithm to perform the combinatorial search in algorithm 1 for arbitrary datasets (Goel et al., 2020). However, this does not preclude the existence of heuristics that tend to work well on problems encountered in practice. We investigated this by using a greedy local search (GLS) over the graph formed by the zonotope's 1-skeleton. We start by selecting a vertex at random and find its corresponding optimal loss by solving a convex program. We iterate over its neighboring vertices and compute their optimal losses as well. We then move to the neighboring vertex with the lowest loss and repeat the process until we arrive at a vertex with lower loss than its neighbors. We then take that vertex's optimal parameters as our approximation to the global minimization problem. This algorithm is defined in detail in algorithm 2.

We also experimented with some additional heuristics that help the GLS converge faster by reducing the number of convex problems solved at each step. For example, we can greedily move to the first neighboring vertex encountered with a lower loss, which significantly decreases the time per step in the early stages of training. We can further improve this by using geometric information about a solution's relative location in its activation region to try certain vertices first. We call the algorithm with these heuristics modified greedy local search (mGLS) and define it in detail in appendix G.

Note that these heuristics are not guaranteed to return a global minimizer of the loss. Furthermore, there are an exponential number of vertices in the zonotope, so there are no immediate guarantees of them taking less than exponential time to run. However, each step takes polynomial time since each vertex has $O(mN)$ neighbors, so each step solves a polynomial number of convex optimization problems.

We ran experiments comparing these heuristics to gradient descent on toy datasets generated in the same way as in section 4.2. We used GLS on the synthetic data and mGLS on the MNIST and Fashion MNIST derived data. We present some of our results in fig. 3. See appendix H for details of the training procedures and for results for more values of $d, m_{\text{gen}}$ and $d, N$ pairs. On synthetic data, the GLS heuristics significantly outperformed gradient descent. On binary classification tasks, mGLS outperformed gradient descent for networks with moderate levels of underparameterization and performed similarly otherwise. We observed a similar trend across the rest of the $d, m_{\text{gen}}$ and $d, N$ pairs. These results support our hypothesis that gradient descent is suboptimal in the combinatorial search over activation patterns. Furthermore, they suggest that this combinatorial optimization might tend to be tractable in practice.

# 5    Related Work

Some of the concepts in this work also arise in Zhang et al. (2018). A key difference is that they analyze the activation regions in input space given a ReLU network with fixed parameters. We can, in fact, use their tropical geometric approach to derive our zonotope formalism for a single ReLU unit. To do so, the roles of the weights and data must be swapped; we instead use a fixed data matrix and varying vector of weights while they use a fixed weight matrix and varying vector of data. Our use of a zonotope generated by the training examples, however, is novel. Misiakos et al. (2021) show that the approximation error between two shallow ReLU networks depends on the the Hausdorff distance between the zonotopes generated by the each network's units. Bach (2017) also use a Hausdorff distance between zonotopes in the context of neural network optimization.

Goel et al. (2020) provide proofs of the NP-hardness of optimization of shallow ReLU networks and the hardness of even finding an approximate solution. Froese et al. (2021) extend these results and show that the brute force search in Arora et al. (2016) cannot be avoided in the worst case. Du et al. (2018) was one of the first works to prove that overparameterization in shallow ReLU networks allows gradient descent to converge to a global optimum in polynomial time. Their bound of $\Omega(N^6)$ on the number of hidden units needed for convergence was improved upon by subsequent work (Ji & Telgarsky, 2019; Daniely, 2019). For example, Oymak & Soltanolkotabi (2020) proved a bound of $\Omega(N^2/d)$.

Pilanci & Ergen (2020) and Wang et al. (2021) represent global optimization of shallow ReLU networks with $\ell_2$ regularization using a convex optimization problem that operates simultaneously over *all* activation patterns for a single unit. Multiple units are handled by summing over the activations with different activation patterns. This leads to exponential complexity in the data dimension $d$ but avoids exponential complexity in the number of units $m$. Dey et al. (2020) provide an example of a heuristic algorithm that searches over activation patterns for a single ReLU unit. Their algorithm operates on the principle that examples with large positive labels are more likely to belong to the active set in good solutions.

# 6    Conclusion

We introduced a novel characterization of the combinatorial structure of activation patterns implicit in the optimization of shallow ReLU networks. We showed that it can be described as a Cartesian product of zonotopes generated by the training examples. We used this zonotope formalism to explore aspects of the optimization of shallow ReLU networks. It provides a natural way to describe instabilities of the global minimum to perturbations of the dataset. We then related this to work on the NP-hardness of global ReLU optimization. In particular, we demonstrated that this optimization problem is still NP-hard even when restricted to datasets in general position, which is commonly assumed of data in practice.

We then explored how combinatorial considerations play into the relationship between overparameterization and polynomial-time optimization of shallow ReLU networks. Namely we interpret known results for gradient descent as stating that a randomly chosen zonotope vertex will be close to one whose activation region contains a good local optimum. We then provide empirical evidence that sufficient overparameterization makes it highly likely that a randomly chosen vertex has a good local optimum. We also provide a polynomial-time algorithm that can find a vertex containing the global optimum using approximately twice the minimum number of hidden units needed to fit the dataset exactly. Finally, we provide a GLS heuristic over zonotope vertices that outperforms gradient descent on some toy problems.

In future work we plan to theoretically and empirically explore heuristics and algorithms that perform well on real-world datasets. We hope to analyze how vertex choice impacts generalization. Further insights might be derived by exploring the connections of hyperplane arrangements to tropical geometry and oriented matroids (Stanley et al., 2004; Oxley, 2006; Maclagan & Sturmfels, 2015). One caveat of our theory is that it applies to only a shallow ReLU network. However, the concepts of activation patterns are still meaningful for deep ReLU networks but require real algebraic geometry for analysis (Basu, 2014; Bochnak et al., 2013). We hope that further research along these avenues will deepen our understanding of neural network training and enable improvements to training in practice.

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
