# OpenReview forum: "A Combinatorial Perspective on the Optimization of Shallow ReLU Networks"
_NeurIPS.cc/2022/Conference — NeurIPS 2022 Accept_

### Official Review · Reviewer_7Q5f · 2022-06-23

**Rating:** 7
**Confidence:** 4
**Soundness:** 3 good
**Presentation:** 3 good
**Contribution:** 3 good

**Summary:**

This paper considers the problem of optimizing a shallow ReLU network. The authors characterize the combinatorial structure of this problem via a zonotope whose vertices correspond to the feasible activation patterns of the network for a given training dataset. They use this characterization to provide some new insights and results on shallow ReLU optimization. In particular, they show that a necessary condition for the global optimum of a shallow ReLu network to be discontinuous is for the training dataset to have non-trivial affine dependencies. This implies that existing NP-hardness results do not trivially apply to datasets in general positions (i.e., no non-trivial affine dependencies between training examples). They also explore the relationship between over-parameterization and optimization difficulty. They provide empirical evidence that over-parameterization increases the likelihood that a randomly chosen vertex will contain a good solution. They propose a polynomial-time vertex selection algorithm which finds the global optimum using only $\Omega(2 N)$ parameters, which improves over existing bounds on the amount of over-parameterization needed. They further introduced a greedy local search heuristic which outperforms gradient descent on  some toy datasets.

**Questions:**

Questions:
- Why no results on real-world datasets are provided? Is the proposed heuristic already too expensive for datasets like CIFAR-10?
- Why do you use GLS on the synthetic data and mGLS on the MNIST and Fashion MNIST derived data, and not mGLS on both?
- What is meant by "ill-conditioned" on line 299? it is helpful to include some empirical results to illustrate this point.

Suggestions:
- Include timing results in the comparison of gradient descent and mGLS.
- Add a comparison with the heuristic algorithm of Dey et al. (2020).
- Change the x-axis of the Fashion MNIST to over-parameterization factor, in both Figures 2 and 3.
- Paragraph on lines 202-209 is not very clear. It would be good to provide further details with formal mathematical descriptions of the two cases.
- Typo in Eq. (3): $2 (a_i - 1) \rightarrow (2 a_i - 1)$

**Limitations:**

The authors discussed limitations of their work in the conclusion. There's no direct negative societal impact as this work is mostly theoretical.

**Strengths And Weaknesses:**

Strengths:
- Novel characterization of the optimization of shallow ReLU networks which enabled the authors to provide interesting and novel results which further our understanding of neural networks training, and can be useful for future research.
- Paper is well written with results clearly presented overall (the clarity of some sections can be further improved, see Questions for details)
- A helpful overview of the mathematical concepts used is provided in the appendix.

Weaknesses:
- Results only apply to shallow ReLU networks.
- The proposed greedy local search heuristic is too expensive to be used in practice on large real-world datasets.
- Empirical results only on toy datasets, and no comparison with other existing works.

---

> ### Author Response · Authors · 2022-08-02
> **Response to Reviewer 7Q5f**
>
> Thank you for your review. The current implementation of our heuristics is not well-suited for full versions of real world datasets yet. Significant gains could come from better utilization of constrained convex solving algorithms and implementations. Additionally, our optimizer focuses on a batch optimization setting rather than a minibatch setting, which negatively impacts its ability to scale to larger datasets. However, we do test our optimizers on smaller versions of MNIST and FashionMNIST to get a better idea of its performance on real-world datasets.
>
> We will update our paper to also provide results of mGLS on the synthetic data; there was no technical reason why we didn’t do this originally.
>
> By “ill-conditioned” on line 299, we mean that the network has a poor condition number as a function. Essentially, the magnitude of the network’s gradient with respect to its input can become quite large. We will provide an example in the appendix of a dataset that can be fit with a well-conditioned ReLU network but for which the algorithm can produce a poorly conditioned network.
>
> The optimizer of Dey et al. (2020) is a bit different from ours in that it only works in the case of a single ReLU unit. Since we focus on multi-unit networks, a comparison with their method would be rather limited.
>
> In the synthetic data experiments, the dataset was labeled by a random ReLU network whose number of hidden units we know, so we knew the minimum number of units needed to fit the dataset. Hence the overparameterization factor was well-defined there. An overparameterization factor would be a bit ill-defined for the MNIST/FashionMNIST datasets since they were not labeled by a ReLU network of known size.
>
> We will add explicit examples in the appendix illustrating the two cases in lines 202-209. Thank you for spotting the typo in equation 3.

---

> > ### Comment · Reviewer_7Q5f · 2022-08-08
> > **Thanks for addressing my questions.**
> >
> > The authors addressed my concerns. Adding the mentioned examples will improve the clarity of the paper. I also agree with Reviewer 6i9S
> >  that stating the theoretical results clearly as theorems would further improve clarity. The additional result discussed in the response to Reviewer Akn8 also significantly strengthen the contribution of the paper.

---

### Official Review · Reviewer_6i9S · 2022-07-08

**Rating:** 5
**Confidence:** 3
**Soundness:** 3 good
**Presentation:** 2 fair
**Contribution:** 3 good

**Summary:**

This paper illustrate that the global optimization of a two-layer ReLU network can be characterized as a combinatorial search over the vertices of the zonotope which is isometric to all feasible activation patterns. This perspective is utilized to show that the global optimum of two-layer ReLU network is discontinuous with respect to the training data. They also interpret the overparametrization as the random sampling over the zonotope vertices. A novel heuristic algorithm of greedy local search over the zonotope vertices is introduced and numerical results show its advantage over the gradient descent method.


**Questions:**

The authors shall formalize their main results in the theorem or proposition environment.

For Figure 3, as the proposed greedy local search heuristic is a randomized algorithm and the initialization of the gradient descent is at random, the authors shall plot the heat map of multiple trials instead of showing the result for a single trial. For instance, in the lower part of the figure 3, by increasing the number of units from 16 to 32, the final training accuracy of mGLS decreases. This may result from the random seed.

Some literature are missing. The concept of zonotope is utilized in the optimization of two-layer ReLU network in [1,2]. The global optimization of the training problem two-layer ReLU networks with the standard $\ell_2$ regularization can also be formalized into the search over all possible activation patterns, see [3,4].

[1] Bach, Francis. "Breaking the curse of dimensionality with convex neural networks."
[2] Misiakos, Panagiotis, et al. "Neural Network Approximation based on Hausdorff distance of Tropical Zonotopes."
[3] Pilanci, Mert, and Tolga Ergen. "Neural networks are convex regularizers: Exact polynomial-time convex optimization formulations for two-layer networks.”
[4] Wang, Yifei, Jonathan Lacotte, and Mert Pilanci. "The Hidden Convex Optimization Landscape of Two-Layer ReLU Neural Networks: an Exact Characterization of the Optimal Solutions."


**Limitations:**

Yes.

**Strengths And Weaknesses:**

Strength:
- The idea of connecting the optimization of ReLU networks with vertices in the zonotope is very interesting.
- The numerical comparison between the greedy local search heuristic and gradient descent algorithm illustrate the strength of the proposed heuristic algorithm.

Weakness:
- This is a theoretical paper but the theoretical contribution of this paper is hard to interpret. There are no formalized theorem or proposition in the paper. It will be very helpful if the author can organize their main contributions into theorems.
- The authors show that when the training data is in general position, the global optimum of the training set is continuous with respect to the training data. But what is the implication of this result?
- The discussion of over-parameterization in section 4.2 only provides a heuristic that the random vertex selection can work and this is only numerically verified. Some solid analysis about why the random vertex selection works will greatly strengthen the paper.

---

> ### Author Response · Authors · 2022-08-02
> **Response to Reviewer 6i9S**
>
> Thank you for your review. The main theoretical results of our paper are:
> - The combinatorial structure of the activation regions of a ReLU network is captured by the zonotope $\mathcal Z^m$. In particular, its set of vertices is in a 1-to-1 correspondence with the set of activation regions.
> - The global minimum of the loss of a shallow ReLU network on this dataset is continuous with respect to the dataset if the dataset is in general position.
> - Optimization of a shallow ReLU network is NP-hard even when the dataset is in general position. Coupled with our continuity proof, this means that NP-hard datasets are not a measure 0 phenomena.
> - Any dataset comprised of N examples belonging to $\mathbb R^d$ can be fit exactly by a shallow ReLU network with $2N/(d+1)$ hidden units in polynomial time.
> We will update our paper to state these explicitly as theorems.
>
> The implications of the global optimum of the training set being continuous with respect to the training data when the data are in general position relates to our results on NP-hard shallow ReLU optimization problems. Coupled with our demonstration of an explicit discontinuity of the global minimum for a dataset that encodes the set cover problem in Appendix D.2, this shows that the NP-hardness of shallow ReLU optimization over datasets in general position does not follow automatically from continuity. Since previous works reduced NP-hard problems to datasets not in general position, this shows that the complexity of optimization over datasets in general position, which is a common assumption for real-world datasets, was an open problem at the time of submission. Since then, we have produced an example of a dataset in general position encoding the set cover problem. See our revised version of Appendix D.3 and our response to Reviewer Akn8 for more information on this.
>
> We briefly mention connections of the random vertex method to random feature models but didn’t go into more detail. We will go into more detail about this, which shows why the random vertex method works as the number of units increases. Essentially, as the number of units $m$ increases past the number of examples $N$, it becomes possible for the set of outputs of the first layer across the training dataset to become linearly independent. The precise conditions for an activation region to contain parameters that produce linearly independent activations can be shown to depend entirely on its activation pattern when the dataset is in general position. If we co-fit the second layer weights, we can then do linear regression over these random features to get zero training loss. In the setting of frozen second layer weights and scalar network output, we can get a similar effect by varying the magnitude of the first layer weights with a cost of roughly doubling the number of units to take into account positive and negative coefficients in the linear regression.
>
> The results presented for Figures 2 and 3 are actually the median of 16 runs for the random vertex experiments and the median of 8 runs for (m)GLS and gradient descent. The results for these experiments along with their standard deviation across runs are presented in Tables A.1 and A.2 in the Appendix. Section G in the Appendix provides detailed information about these experiments. We will make the fact that the results presented in Figures 2 and 3 are aggregates across multiple runs more clear in the main text. We will also add some indicator of inter-run variance to the figures themselves either via error bars or a heatmap.
>
> Thanks for bringing up the missing literature. Our construction and use of a zonotope differs from [1,2], but they are important to mention as prior work using zonotopes in the optimization of shallow ReLU networks. The approaches taken by [3,4] are very interesting! They are certainly related to optimizing over activation patterns, and we will mention them in our related work. We should mention that our methods can also easily be adapted to work with convex regularizers on the weights since they do not affect the activation regions.

---

> > ### Comment · Reviewer_6i9S · 2022-08-04
> > **Response to the authors**
> >
> > The authors address most of my questions and I raise the score.

---

### Official Review · Reviewer_Akn8 · 2022-07-08

**Rating:** 5
**Confidence:** 4
**Soundness:** 3 good
**Presentation:** 3 good
**Contribution:** 2 fair

**Summary:**


This paper uses methods from polyhedral combinatorics to study the optimization landscape of Shallow ReLU networks.
By associating a proper combinatorial object (zenotope) with an instance of minimizing the training error of a ReLU the authors show
that hardness results may be non robust and might break under small perturbations. Another contribution is to use the zenotope
to introduce a local search algorithm and demonstrate experimentally that it can outperform local search.

**Questions:**

If I recall correctly, the paper of Vu regarding the hardness of training shallow networks argues that the hardness results there should persist even if the training points are in general positions. Doesn't this indicate that the optimization problem for perturbed data sets should be hard as well? Personally I think it would be surprising (though not impossible) if the hardness results for training shallow networks would break under (small amounts of) noise.

**Limitations:**

A lack of a proof of a new fact of significance.

**Strengths And Weaknesses:**

I like the idea of analyzing hard instances arising from NP hardness proofs. However the paper falls short to prove or disprove whether
NP hardness results are robust in the sense that they may hold for non-degenerate data sets (that can be obtained by random perturbations). Without a resolution of this I find the contribution of the paper to be below the bar of NeurIPS.

---

> ### Author Response · Authors · 2022-08-02
> **Response to Reviewer Akn8**
>
> Thank you for your review. Do you have a reference for the paper by Vu? It is not among our references and a Google search failed to bring up any results. To the best of our knowledge, we have been unable to find a paper making such claims.
>
> After submission of this paper, we managed to prove that the modification of Goel et. al (2020)’s reduction of the set cover problem that we presented in Appendix D.3 is indeed a reduction of the set cover problem whose dataset is in general position. We have uploaded a version of the Appendix D.3 with the new proof in the revised supplemental material.
>
> Here is an overview of the proof. We needed to show that if the minimal loss of the perturbed dataset is greater than $t  \gamma^2 / N$, then no set cover of size $t$ exists. We start by showing that if the minimal loss over the original dataset is equal to or less than $t  \gamma^2 / N$, then the minimal loss over the perturbed dataset is equal to or less than $t * \gamma^2 / N$ as well. The contraposition of this statement is that if the minimal loss over the perturbed dataset is greater than $t  \gamma^2 / N$, then the minimal loss over the original dataset must be greater than $t  \gamma^2 / N$. As proved in Goel et. al (2020), the latter statement implies that no set cover of size $t$ exists. Hence if the minimal loss over the perturbed dataset is greater than $t  \gamma^2 / N$, then no set cover of size $t$ exists.
>
> Coupled with our proof that the global optimum of the loss is continuous when the dataset is in general position, our construction implies that NP-hard datasets are not a measure 0 phenomenon, i.e. they form a set of positive Lebesgue measure when considered as a subset of $\mathbb R^{N \times d}$.
>
> We believe that adding this proof addresses your concern about the lack of a proof of a novel fact of significance. As we have shown with our proof on the continuity of the global optimum and demonstration of its discontinuity for a dataset used in an NP-hardness proof, the NP-hardness of optimizing a shallow ReLU over a dataset in general position was an open problem before this work. Since most real-world datasets are assumed to be in general position, the complexity of optimizing over such datasets has relevance to practice. To the best of our knowledge, we are first to prove this problem to be NP-hard for datasets in general position.

---

### Official Review · Reviewer_L3JB · 2022-07-11

**Rating:** 6
**Confidence:** 2
**Soundness:** 3 good
**Presentation:** 3 good
**Contribution:** 3 good

**Summary:**

This paper presents a combinatorial view of optimizing shallow (two-layer) ReLU networks. The authors show that optimizing a shallow ReLU network is an NP-hard combinatorial optimization problem, and existing optimizers such as gradient descent are performing some local combinatorial search. The authors then present a greedy search method to exploit the combinatorial optimization nature, and it outperforms gradient descent in experiments.

**Questions:**

1. Is it possible to compare other optimizers like SGD?

**Limitations:**

The limitations are addressed.

**Strengths And Weaknesses:**

**Strengths**
1. The combinatorial modeling perspective of the learning process of shallow ReLU networks is novel and interesting.
2. The theoretical analysis and insights presented in this paper seem sound. For example, the authors present explains the relationship between the NP-hard difficulty found in this paper and the recent theoretical results that optimizing overparameterized deep networks is feasible in polynomial time.
3. Empirical results are provided to ground the theoretical findings. The proposed greedy search method for shallow ReLU networks is interesting.

**Weaknesses**
See "Questions"

---

> ### Author Response · Authors · 2022-08-02
> **Response to Reviewer L3JB**
>
> Thank you for reviewing our paper. To answer your question on comparing to other optimizers like SGD, the novel optimizers introduced in our paper operate in the batch setting where optimization occurs over the entire training dataset at once. Since we use the entire training dataset to construct the zonotope, our methods do not have an immediate translation to the mini-batch setting. That is why we only compared to batch gradient descent in this work. We hope to develop similar optimizers that operate in the minibatch setting in future work.
>
> We could also compare to batch versions of other optimizers used in practice such as Adam, but we focused on using vanilla gradient descent due to its simplicity. Previous theoretical work on ReLU optimization also usually focused only on vanilla gradient descent as well.

---

> > ### Comment · Reviewer_L3JB · 2022-08-09
> > **Thanks for your response.**
> >
> > Thanks for your response. I retain my original score (6: Weak Accept).

---

### Meta-Review · Area_Chair_2hS3 · 2022-08-23

**Recommendation:** Accept
**Confidence:** Certain

**Metareview:**

Thank you for your submission to NeurIPS.

This work presents a combinatorial view of training two-layer ReLU networks.

The reviewers and I, after the author response, are in agreement that there are interesting and strong contributions in this work. Namely, it is shown that global optimization of a two-layer ReLU network can be characterized as a combinatorial search over the vertices of a zonotope. This is used to show that the global optimum of two-layer ReLU network is discontinuous with respect to the training data. Four knowledgeable reviewers recommend accept/borderline accept, and I concur, in light of the contributions made.

The reviewers also noted some minor weaknesses: In particular, the reviewers noted that
(1) results only apply to two-layer ReLU networks
(2) proposed greedy local search heuristic is too expensive to be applied on modern datasets.


Please take into account the updated reviewer comments, including suggested additional references, when preparing the final version to accommodate the requested changes.





**Award:**

No

---

### Decision · Program_Chairs · 2022-09-14

Accept